# HIT-GCN: Spatial-Temporal Graph Convolutional Network Embedded with Heterogeneous Information of Road Network for Traffic Forecasting

**Haitao Xiong, Guojiang Shen, Xiang Lan, Haopeng Yuan and Xiangjie Kong ***

College of Computer Science & Technology, Zhejiang University of Technology, Hangzhou 310023, China
* Correspondence: xjkong@ieee.org

**Abstract:** In road networks, attribute information carried by road segment nodes, such as weather and points of interest (POI), exhibit strong heterogeneity and often involve one-to-many or many-to-one relationships. However, research on such heterogeneity in traffic prediction is relatively limited. Our research examines how varying the network propagation pattern based on the degree of node-to-node heterogeneity of information affects the model prediction performance. Specifically, at the node level, we use knowledge embedding to generate knowledge vectors that quantify the heterogeneity among the attribute information of a node. At the road network level, we calculate a homogeneity adjacency matrix that captures both the topological structure of the road network and the similarity of node heterogeneity. This adjacency matrix assigns different weights to neighbors based on their homogeneity, guiding the propagation of graph convolutional networks (GCN). Finally, we separate the representation of propagation into self-representation and neighbor representation to extract multi-attribute information, including self, homogeneity, and heterogeneity. Experiments on real datasets demonstrate that the incorporation of our homogeneity adjacency matrix leads to a significant improvement in the accuracy of short-term and long-term prediction compared with previous work on homogeneous and single-dimensional information. Furthermore, our approach maintains its performance advantage over baseline models under different embedding dimensions and parameter settings.

**Keywords:** traffic forecasting; heterogeneity information; adjacency matrix; knowledge graph embedding



## 1. Introduction

Transportation has a significant impact on the operation of modern society and the development of production, as well as deeply affecting the daily lives of individuals. With the increasing number of vehicles and the complexity of road networks, transportation problems have become more acute, especially in urban areas, where urgent solutions are needed. To meet the traffic control requirements of cities, an intelligent transportation system capable of predicting, analyzing, and controlling traffic is needed to create a safe and orderly urban transportation management [1]. Traffic speed prediction is a key task in traffic management and planning. Successful traffic speed prediction can enable a series of applications, such as real-time traffic management, route planning, and intelligent transportation systems, which ultimately contribute to improving the safety and efficiency of traffic networks. Traffic speed is defined as the average speed of vehicles passing through a spatial unit during a given period of time. The speed value on urban roads can reflect the degree of congestion in road traffic, which can also serve as a reference index for applications such as route navigation and arrival estimation. Traffic speed prediction aims to predict the speed of future traffic flow given a road network and historical traffic observations (e.g., recorded by sensors). Traffic speed prediction plays an important role in

urban traffic planning, management, and control, providing decision support for urban vehicles. For example, traffic speed prediction can help vehicles make optimal decisions by predicting the speed of traffic flow on a road section during a certain period of time, thus obtaining the fastest route and avoiding traffic congestion.

However, traffic speed is affected by various external factors, such as weather conditions, bus stops, emergency events, holidays, and the distribution of nearby points of interest (POIs). These external factors directly or indirectly affect traffic speed prediction. For example, changes in weather conditions may affect road conditions and, in turn, traffic speed [2]. The distribution of POIs may also indirectly affect traffic speed [3]. Typically, in areas with dense POI distribution, traffic flow is greater, and traffic speed is relatively slower. Meanwhile, these external information sources exhibit strong heterogeneity, with multiple-to-one or one-to-many relationships existing between road segment nodes and attribute information, such as a single road segment corresponding to multiple POIs or a weather data point corresponding to multiple road segments and dates. Therefore, how to quantify the heterogeneity of information and then utilize this heterogeneity in our research work has become a major focus of our study.

In knowledge graphs, networks with different types of objects and relations are referred to as heterogeneous information networks [4]. We consider external factors that influence urban traffic prediction as heterogeneous external information (such as weather, POI distribution, etc.), because the entities, attributes, and relationships of this information are all heterogeneous. At the level of individual nodes in the road network, we embed the heterogeneous information of the nodes, so that the resulting embedding vectors can contain this heterogeneous information, and different attributes and relationships can be quantified. In the overall network framework, the degree of heterogeneity between the attribute information carried by nodes is compared to determine whether two road nodes have similar spatial characteristics, fully utilizing the differences in this information to better assist us in traffic prediction and improve prediction accuracy and reliability. Existing research that considers external factors solely focuses on these factors, ignoring the influence of the relationship between traffic information and external factors on traffic. A few methods have utilized this information [5], but have not fully leveraged the heterogeneity of this information. We use knowledge graph embedding to quantify this external heterogeneous information and thus utilize the heterogeneity of this attribute information.Our work makes the following contributions:

1.  We propose a framework for processing spatiotemporal graph convolutional networks with heterogeneous information, which addresses the heterogeneity problem among attribute information carried by road network nodes in the field of traffic prediction. Compared with previous work that dealt with homogeneous information and single-dimensional information, we achieve good results in processing heterogeneous information.
2.  We calculate a homogeneity adjacency matrix based on the topological structure of the road network and the similarity degree of information heterogeneity between nodes. This guides changes to the propagation method of GCN, allowing it to better account for the node characteristics of "heterogeneous information and topological structure" that are similar. At the same time, we divide the feature representation into self-representation and neighbor representation to facilitate the extraction of multi-attribute information (self, homogeneity, heterogeneity).
3.  Finally, we conduct comparative and ablation experiments on two real datasets of Shenzhen taxis and the corresponding heterogeneous information possessed by road sections, and verify the excellent performance of the model.

The rest of this paper is organized as follows. In Section 2, the related work of spatiotemporal graph convolutional networks and knowledge representation is provided. Section 3 discusses methodology and formulation settings. The viability of the strategy is then empirically proven in Section 4. Section 5 is finished with a summary.

## 2. Related Work

### 2.1. Spatial-Temporal Graph Convolutional Network

In this era of the Internet of Everything (IoE), the data collection and application of large-scale interconnected devices have solved many practical problems [6]. In the field of the Internet of Vehicles, especially, the role is significant. Gaohui Duan et al. designed a joint computing and caching framework by integrating a deep deterministic policy gradient (DDPG) algorithm to solve the energy cost problem that is significant to the Mobile network operators [7]. Xiangjie Kong et al. utilized vehicle trajectory data generation model [8] and analyzed urban division and mobility patterns. Interconnected speed collection devices at road junctions provide powerful data support for traffic flow prediction. In the field of traffic flow forecasting, traditional forecasting models, such as the History Average Model and the Time-Series Model, usually use statistical analysis to predict traffic conditions. With the continuous development of machine learning and deep learning, the application of deep neural network models in the field of traffic prediction is gradually mature. Tedjopurnomo, David Alexander, et al. explained in detail the popular deep neural network architecture commonly used in traffic flow prediction literature, and summarized the similarities and differences between the various research [9]. Convolutional neural networks and recurrent neural networks (RNNs) [10] are also used in traffic prediction problems to model temporal and spatial correlations. Long short-term memory (LSTM) and gated recurrent units (GRU) [11], which are frequently used to explore the temporal characteristics of traffic data, are among them. However, the general deep learning model can only be applied to Euclidean data (i.e., image, text and video). When processing traffic data, the road network data needs to be converted into grid data, which destroys the inherent connectivity of the road network. Additionally, graph neural networks (GNNs) have developed to effectively address this weakness because graph structure is naturally suitable for the modeling of road networks. GNNs have recently taken the lead in deep learning research [12], demonstrating cutting-edge performance in a range of applications [13]. As a neural network that operates directly on graph structures, GNNs can recognize intricate relationships between objects and conclude graphically represented data. GNNs have been proven to be effective in various node-level, edge-level, and graph-level prediction tasks [14]. In addition, GNNs may be roughly divided into four types: recursive GNNs, convolutional GNNs, graph autoencoders [15], and spatio-temporal GNNs [16]. On the type of convolutional GNN, a neural network model of graph convolution, GCN, was gradually developed [17]. GCN is a convolution GNN based on spectrum, in which graph convolution is defined by introducing a filter from graph signal processing in the spectrum domain (e.g., Fourier domain).

Many researchers have used GNNs in the study of traffic because of the model characteristics of GCNs. Numerous variations of GNNs have been used in the analysis of traffic as a result of the development of GCNs. To address the congestion recognition problem, Shen, Guojiang, et al. proposed an attention-based directed graph convolutional network (ADGCN)-supported framework [18]. Xiangjie Kong et al. proposed a dynamic graph convolutional recurrent imputation network (DGCRIN) [19], which employs a graph generator and dynamic graph convolutional gated recurrent unit (DGCGRU) to perform fine-grained modeling of the dynamic spatiotemporal dependencies of road network. Zhi Liu et al. proposed a dynamic multi-view coupled graph convolution (DMV-GCN) [20] to solve the problem that urban travel demand is affected by regional functions such as weather. Liang Yang et al. proposed a new graph convolution network (TO-GCN) [21] based on topology optimization, which fully exploits the potential information by simultaneously improving the network topology and learning the Topology Optimization based parameters of the fully connected network (FCN). Shengnan Guo et al. proposed the ASTGCN [22] model, which employs both graph convolution and attention mechanisms to model traffic data in the form of network structures. Ling Zhao et al. proposed the temporal graph convolution network model (T-GCN) [23] to capture both spatial and temporal dependencies, which fuses graph convolution and gated cyclic units. Jiawei Zhu proposed the KST-GCN [24] model, which captured the spatio-temporal characteristics of traffic while capturing the knowledge

structure and semantic relationships between traffic information and attributes. Using a data-driven approach, Ping Lu et al. proposed a dynamic spatio-temporal perceptual graph to replace the predefined static graph in traditional graph convolution. Based on this, they also designed a new graph neural network architecture DST-GCN [25]. The model not only represents dynamic spatial correlations between nodes through an improved multi-headed attention mechanism but also obtains a wide range of dynamic temporal dependencies from multiple perceptual field features through multi-scale gated convolution. To learn human mobility knowledge from the fixed human mobility patterns. Xiangjie Kong et al. proposed a multi-pattern passenger flow prediction framework, MPGCN [26], based on GCN, to address this research gap.

### 2.2. Knowledge Representation and Knowledge Embedding

In GCN, the adjacency matrix is seen as the key to capturing spatial correlation in traffic prediction. We were motivated by the work mentioned above, where they directly or indirectly altered the adjacency matrix and consequently influenced the propagation method to obtain a more suitable architecture. To quantify the heterogeneous information of road segments, we create a knowledge graph by embedding the heterogeneous information of nodes in the road network using knowledge mapping techniques. The basic composition of a knowledge graph is a triad of <entity, relation, entity>, and entities are linked to each other by relations, forming a complex mesh of knowledge structures [27], which is mostly used for modeling relational data. In recent years, knowledge graphs have been widely used in applications such as recommender systems [28–31], finance [32,33] and e-commerce [34]. We use knowledge graph techniques to embed heterogeneous information about nodes in a road network and model the data for heterogeneous information. The core idea behind Knowledge Representation and Knowledge Graph Embedding (KGE) [35] is Learning from the representation of things and relations in a knowledge base. By projecting entities or relationships onto low-dimensional vector space, semantic information representation and embedding of entities and relationships are realized. To achieve knowledge representation and embedding, Bordes' TransE [36] approach proposed in 2013 interprets relationships in a knowledge base as a type of translation vector between entities. TransH [37], which is based on TransE, proposes employing hyperplane normal vectors and translation vectors to express relationships to overcome the complex relationship representation problem. Yankai Lin et al. proposed TransR [38] and KR-EAR [39] based on both TransE and TransH, which separate KG relations into attributes and relations. They use attribute-based tuple encoding to embed the relations between entities and attributes, and associative tuple encoding to embed the links between entities and relations, acting as a classification model in attribute prediction. The model can depict both the properties shared by entities as well as their relationships. We need a model that can retain a more extensive range of information and structure because external data on urban transportation (such as weather and POI) is made up of numerous diverse entities, relationships, and attributes, which is a rich type of data with complex interactions. As a result, we select the KR-EAR model as our knowledge representation model.

## 3. Methods

### 3.1. Framework and Problem Definition

The goal of traffic speed forecasting is to predict traffic speed in a specified period through previous road traffic speeds as well as some external data. In the original network for predicting traffic speed, its input is only the current speed characteristics of the road section, the information of the nodes and the heterogeneity between them is not taken into account. Based on these, spatial-temporal graph convolutional network embedded with heterogeneous information of road network(HIT-GCN) is proposed, as shown in Figure 1. We first fuse the multi-attribute external heterogeneous information and the original velocity features to obtain the knowledge graph embedding vector, which is used as the input of our GCN network. At the same time, we calculate the embedding correlation matrix

from the obtained embedding vector and combine it with the original network topology adjacency matrix with adjustable parameters to obtain the final homogeneous adjacency matrix. Finally, this homogeneous adjacency matrix is fed into the spatio-temporal convolutional network together with the constructed knowledge graph embedding vector for the extraction of spatio-temporal features to obtain the final speed forecasting.

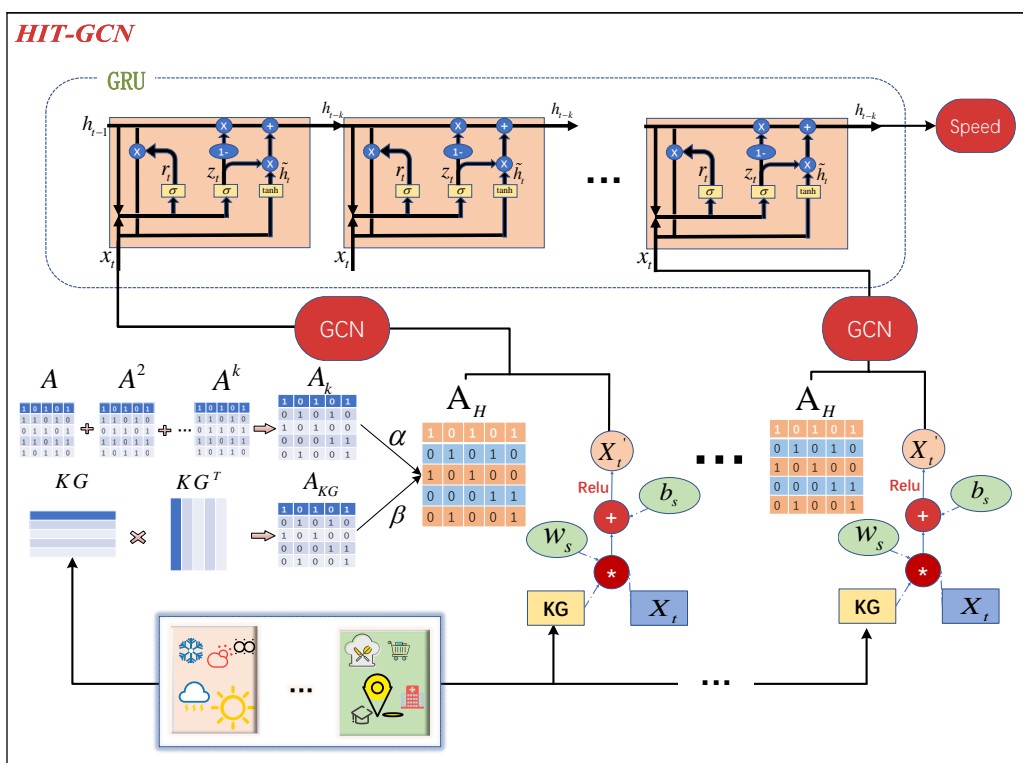

**Figure 1.** The framework of the proposed HIT-GCN.

In ITS, we define each road segment as a node, and the connection relationship between road segments as the edges connected between nodes, so that we map the road network into a graph structure, represented $G = (V, E)$, where $V = \{v_1, v_2, \ldots, v_n\}$ is the set of nodes, $v_i$ denotes the ith road and n is the number of roads and $E = \{e_{i,j}\} \subseteq V \times V$ is the set of edges connecting roads with $e_{ij} = 1$ indicates that road section *i* is connected to section *j*, Otherwise $e_{ij} = 0$ means not connected.

Following the definition of the roadmap network *G*, we combine the knowledge graph *KG* made up of external heterogeneous information, the speed feature matrix *X*, and the roadmap network *G* into a learning function $y = f(G, X, KG)$, In other words, using historical and future velocity information, as well as external heterogeneous information on the road segment nodes and topology, as stated in Equation (1), we learn the velocity information for the period *T*:

$$[x_{t+1}, x_{t+2}, \ldots, x_{t+T}] = f(G, X, KG) \tag{1}$$

where *X* is the characteristic matrix of the road network, $x_i^t$ represents the speed attribute of the road section *i* at moment *t*, $KG_i^t$ in the knowledge graph denotes the heterogeneous information attributes external to road segment *i* that affect traffic speed at moment *t*.

After defining the road network *G* and learning function $y = f(G, X, KG)$, we describe the algorithm for the cellular model called HIT-Cell in heterogeneous information spatio-temporal graph convolutional networks, as shown in Algorithm 1. To accurately predict the traffic speed values, we first implement the computation of a knowledge graph embedding vector that quantifies the heterogeneous information, then compute a homogenous adjacency matrix that is fed into GCN network along with a feature matrix that fuses the

knowledge graph with the original speed features to capture the spatial attributes of the nodes. Finally, we compute a GRU that captures the temporal attributes of the nodes.

---

**Algorithm 1** HIT-Cell of HIT-GCN.

---

**Input:** The node feature $X_t$ of the traffic speed, Weather, POI
1: Calculate the embedding vector $X_E \leftarrow$ Weather, POI using Equation (7);
2: Calculate heterogeneous information similarity matrix $AKG$ using Equation (8);
3: Calculate topological similarity adjacency matrix $T \leftarrow A + A^2 + \cdots + A^k$;
4: Update the homogeneous adjacency matrix $A_H \leftarrow \alpha AKG + \beta T$ using Equation (10);
5: Calculate $X_t' \leftarrow X_t' = Relu(KG * X_t * w_s + b_s)$ using Equation (12);
6: Update $X_t' \leftarrow X_t'^{(l)} = \sigma(\mu Z^{(l-1)}W_e^{(l)} + \xi \hat{D}^{-1}A_H X_t'^{(l-1)}W_n^{(l)})$ using Equation (13)
7: Calculate $y^t, h_t \leftarrow GRU(X_t', h_{t-1})$
**Output:** $\hat{Y}$ denotes the predicted value of traffic speed

---

### 3.2. Embedding Heterogeneous Information

By embedding the Entity and Relationships in the knowledge graph into a continuous vector space, knowledge graph embedding is a crucial method for solving the complementation problem of the knowledge graph. This method preserves the structural data of the knowledge graph while streamlining computation. Traditional knowledge representations from the past frequently contained entities and relationships in a low-dimensional semantic space and made the assumption that all relationships were the same, which is different from the heterogeneous data we need to incorporate. The information that road nodes possess is frequently heterogeneous, and both the attributes and relationships are frequently diverse. For instance, attribute relationships for roads may be many-to-one or one-to-many. We utilize the entity-attribute relationship knowledge representation model KR-EAR to effectively handle the embedding of this heterogeneous information. To accurately capture the attribute and relationship components of the heterogeneous information, this model encodes in three dimensions: entity, relationship, and attribute. In our study, we can effectively exploit the section's heterogeneous information by precisely capturing the heterogeneous attribute associations of various entities. The KR-EAR paradigm uses relational encoding and attributes encoding as its two types of encoding. The embedding of node heterogeneous information will be calculated in the sections that follow in two parts.

It is our goal to describe the relationships between entities in the relational tuple encoding portion, and we define the conditional probability $P$ to be learned as:

$$P((v_i, adj, v_j)|X_{KG}) = \frac{\exp(f_{rel}(v_i, adj, v_j))}{\sum_{v_{i'} \in V} \exp(f_{rel}(v_{i'}, adj, v_j)))} \tag{2}$$

where the triplet $(v_i, adj, v_j)$ represents the connection between node $v_i$ and node $v_j$, $X_{KG}$ is the relational embedding vector that we need to learn, and $f_{rel}(v_i, adj, v_j)$ is a function of the energy of TransR.

$$f_{rel}(v_i, adj, v_j) = - \parallel v_i M_r + adj - v_j M_r \parallel_{L1/L2} + b_1 \tag{3}$$

where $M_r$ is the weight matrix of the TransR function and $b_1$ is the bias term. Of course, other KR models such as TransD [40] KG2E [41], and TranSparse [42] can also be easily trivially encoded for the relation. Since the excellent performance of TransR was verified in the KR-EAR model. In our work, we use TransR as the energy function for this part, and the function is defined as in Equation (3).

In the attribute tuple encoding section, which is then used to model the association of entities and attributes, in the association of attributes section, it is considered that multiple attributes of the same entity are related in some way, and we formulate all attributes as:

$$R\_att = \{(v_i, att_l, att_l\_v_i)\}, i, j \in \{1, 2, \ldots, n\}, l \in \{1, 2, \ldots, L\} \tag{4}$$

where in $(v_i, att_l, att_l\_v_i)$, $att_l$ denotes the lth class attribute of node $v_i$, $att_l\_v_i$ denotes the specific value of the lth class attribute of node $v_i$, We define the conditional probability $P$ to be learned as:

$$P((v_i, att_l, Key_l^i))|X_{KG}) = \frac{\exp(f_{att}(v_i, att_l, Key_l^i))}{\sum_{Key_l^{i'} \in VKey_l^i} \exp(f_{att}(v_i, att_l, Key_l^{i'}))} \tag{5}$$

In this part, the entity embedding is transformed into the attribute space by a single-layer neural network, and then the semantic similarity between the transformed embedding and the embedding of the corresponding attribute value is calculated. Similarly, $f_{att}(v_i, att_l, att_l\_v_i)$ is a function of TransR:

$$f_{att}(v_i, att_l, Key_l^i) = - \parallel f(v_i W_{att_l} + b_{att_l} - T_{att_l\_Key}) \parallel_{L1/L2} + b_2 \tag{6}$$

where $f(\cdot)$ is a nonlinear function, such as tanh, $b_{att_l}$ is the bias term for attribute $att_l$, $T_{att_l\_Key}$ is the embedding of the attribute value $Key_l^i$, and $b_2$ is the bias.

The objective optimization function, which is to maximize the joint conditional probability to achieve the ultimate embedding vector $X_{KG}$ that we require, is then jointly determined by combining the relation tuple and the attribute tuple. This is defined below:

$$\begin{aligned} P(R, R_{att}|X_E) &= P(R|X_E)P(R_{att}|X_E) \\ &= P((v, adj, v_j)|X_E)P((v_i, att_l, att_l\_v_i)|X_E) \end{aligned} \tag{7}$$

In conclusion, we utilize the knowledge representation model KR-EAR in this section of the study to gain the embedding vector from the attribute and relationship dimensions, respectively. This embedding vector quantifies the external attribute and relationship information of the road section entities, effectively fusing the external heterogeneous information of the nodes into the embedding vector. This embedding vector serves as a strong foundation for the computation of the homogeneous adjacency matrix and the fusion of the original velocity features with trustworthy external heterogeneous information.

### 3.3. Homogeneous Adjacency Matrix

#### 3.3.1. Homogeneous Adjacency Matrix Operations

By calculating the similarity of heterogeneous information between nodes and the similarity of topology during the propagation process, the homogeneous adjacency matrix is created to alter the propagation of the original GCN so that it is better able to handle neighbouring nodes with close similarity. The influence of these two dimensions on the characteristics of the nodes is obvious. For this reason, we calculate the final homogeneous adjacency matrix from each of the two dimensions, and finally linearly merge the results of the two dimensions using adjustable parameters, as shown in Figure 2

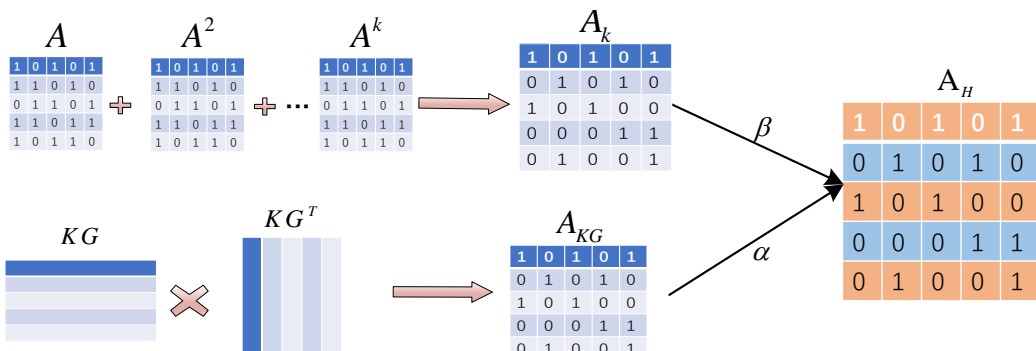

**Figure 2.** Homogeneous adjacency matrix operations.

The first aspect uses the degree of similarity of the heterogeneous information to calculate the neighbourhood weights of two nodes, In the above work the heterogeneity embedding vector $X_{KG}$ which makes each node have some external heterogeneous information about the current section is calculated, so we can use the embedding vector to calculate the heterogeneous correlation between nodes. To express the heterogeneous correlation between nodes, which is partially characterized by utilizing $KG$ instead of $X_{KG}$, we define a heterogeneous information similarity matrix called $AKG$. As seen below,$AKG$ the $AKG$ is calculated:

$$AKG = KG \times KG^T \tag{8}$$

where $KG$ denotes the heterogeneous information embedding representation vector and $AKG_{i,j} = KG_i \times KG_j{}^T$ denotes the degree of similarity between the node $a_i$ and $a_j$ node with heterogeneous information, i.e., the weight between nodes in the propagation process of the network.

The topology of neighbouring nodes and surrounding nodes, on the other hand, is what is meant by the topological properties of a traffic road network. The traffic conditions on the upstream roads have a significant impact on the downstream roads through the state shift in time in the field of traffic, for example, traffic accidents, traffic jams, and other occurrences will directly affect the state of vehicles on the downstream roads. This is based on the characteristics of the traffic flow. Therefore, we obtain the topological similarity adjacency matrix T from the network topology information and perform structure transfer in the k-order network to obtain the network structure of multi-order neighbours. As a result, the importance of the upstream and downstream topology to the traffic road network is obvious, and roads with similar road network topology have a higher similarity. Equation (9) defines the topological similarity adjacency matrix $T$:

$$T = A + A^2 + \cdots + A^k \tag{9}$$

In our work, $k$ was finally adjusted to 3 because the impact of a node's 3rd order neighbours has gradually diminished in earlier studies, we had the best results with $k = 3$ in subsequent trials, and the task's training time was significantly decreased given the algorithm's complexity.

To produce the final homogeneous adjacency matrix A from the two dimensions of network topology similarity and heterogeneous information similarity degree, calculated as follows, we set two learnable hyperparameters to provide the model structure greater dependability:

$$A_H = \alpha AKG + \beta T \tag{10}$$

where $\alpha$ and $\beta$ are hyper-parameters. As $\alpha$ approaches 1, our homogeneous adjacency matrix concentrates more on the heterogeneous correlation between nodes. In a similar vein, as $\beta$ approach 1, the model concentrates more on the network architecture. The following experimental study will illustrate how we have established through thorough testing that the model performs at its peak when $\alpha = 0.4$ and $\beta = 0.8$. Because $T$ reflects the similarity of network architecture between nodes and $AKG$ reflects the similarity of heterogeneous information between two nodes, which do not conflict with one another and do not affect the results of the experiment, the linear combination is possible.

### 3.3.2. Feature Propagation Process of HIT-GCN

Our main aim is to effectively change the propagation path of GCN and the focus of feature extraction so that more useful spatial characteristics can be discovered. We use the resulting homogeneous adjacency matrix to apply graph convolution because the homogeneous adjacency matrix constructed in this method represents the similarity between two nodes with heterogeneous information and the similarity of the network architecture. Feature propagation based on the similarity provided by the homogeneous adjacency matrix and the weights between nodes increases the extraction of features from nodes with the same heterogeneous information and similar network topology while re-

ducing the influence of nodes with heterogeneous information and different attribute. To make sure that the self-information is adequate, we additionally put up a new representation that is split into a self-representation and a neighborhood-representation. While the neighborhood-representation uses the weights given to neighbouring nodes by the homogeneous adjacency matrix, allowing the neighborhood-representation to effectively obtain the feature information of the surrounding nodes, the self-representation uses the output of the upper layer of the current model as input to the current layer. In summary, we give the HI-GCN model of heterogeneous information graph convolutional networks generated based on heterogeneous information attributes as well as network topology, denoted as:

$$Z^{(l)} = \sigma(\mu Z^{(l-1)} W_e^{(l)} + \xi \hat{D}^{-1} A_H Z^{(l-1)} W_n^{(l)}) \tag{11}$$

where $\mu$ and $\xi$ denote the weights of self-representation, and neighborhood-representation respectively, $\hat{D}$ is the diagonal matrix, $Z^{(0)} = X$ represents the attributes of the original node, $\sigma(\cdot)$ and is the activation function.

### 3.4. Homogeneity-Guided Propagation of Spatio-Temporal Graph Convolutional Networks

The main objective of our work is to perform accurate traffic speed forecasting by using the node features of the road network for training purposes, and the node features are classified into spatial and temporal attributes. Therefore, to learn the node features more efficiently, we design the model not only by extracting the spatial features from the road network through GCN but also by considering the learning of the temporal features. Finally, the two parts of the extracted features need to be effectively combined to form a whole network.

#### 3.4.1. Spatial Convolution Module

For the node feature learning process, the usage of knowledge graphs composed of diverse information carried by road network nodes is crucial. In the above process, we calculated the homogeneous neighbourhood matrix from the computed knowledge graph $KG$, which was adopted in the input part of GCN to highlight the strong influence of heterogeneous information on the nodes. An adjacency matrix and a feature matrix make up the output part of a standard GCN network, which enables GCN network to learn the feature properties of the nearby nodes based on the adjacency matrix. In this section, we change the input feature matrix by combining the knowledge graph we created ($KG$) with the initial road segment features $X_t$ to produce the final road segment features $X_t'$, which are computed as follows:

$$X_t' = Relu(KG * X_t * w_s + b_s) \tag{12}$$

Then, to extract spatial features, we input the fused features $X_t'$ and the computed homogeneous adjacency matrix into GCN network. We define a function $gc(X_t'^{(l-1)}, A)$ to represent the full GCN module, which is represented by the following equation:

$$f_{gc}(X_t'^{(l)}, A) = \sigma(\mu X_t'^{(l-1)} W_e^{(l)} + \xi \hat{D}^{-1} A_H X_t'^{(l-1)} W_n^{(l)}) \tag{13}$$

where $W^{(l)}$ is the layer convolution's weight matrix, $X_t'$ is the fused features of the original features and the external heterogeneous information, The first layer of $X_t'$ is the initial value of the fused knowledge graph and section features, and $A_H$ is the homogeneous adjacency matrix containing the heterogeneous information and the network topology.

#### 3.4.2. Temporal Convolution Module

After extracting the spatial feature attributes, we then perform time-dependent capture using recurrent neural networks in the field of time series forecasting, and to finish this study project, we utilize the gated recurrent unit GRU. The GRU was chosen because

of its performance and principles, which are similar to those of the LSTM, as well as its simplicity and ease of training. Additionally, because it has fewer parameters to tweak, the experimental component and training time may be effectively optimized. The following is an example of a GRU formula:

$$
\begin{aligned}
u_t &= \sigma(W_u f_{gc}([X'_t, h_{t-1}], A) + b_u) \\
r_t &= \sigma(W_r f_{gc}([X'_t, h_{t-1}], A) + b_r) \\
c_t &= \tanh(W_c f_{gc}([X'_t, (r_t \odot h_{t-1})], A) + b_c) \\
h_t &= u_t \odot h_{t-1} + (1 - u_t) \odot c_t
\end{aligned}
\tag{14}
$$

where $[\cdot]$ denotes splicing the output of the previous time series with the input of the current layer, $\odot$ denotes the Hadamard product, $r_t$ is the GRU reset gate, and $u_t$ denotes the update gate.

### 3.4.3. Loss Function

After completing this part of the work, we feed the output $h_t$ of the GRU via the fully connected layer to acquire the predicted speed value $\hat{Y}$. We then compare the predicted speed value with the actual speed of the corresponding road segment to reduce the difference between the two values to obtain the most accurate prediction. The loss function is set as:

$$
loss = \| Y - \hat{Y} \| + \lambda L_{reg}
\tag{15}
$$

where $L_{reg}$ is the $L2$ regularisation term to avoid overfitting, the control weights are kept small and $\lambda$ is the hyperparameter of the loss function.

## 4. Experiments

### 4.1. Experiment Setup

#### 4.1.1. Datasets

Due to the specificity of the conditions for building the knowledge graph, we chose the Shenzhen taxi dataset, because in addition to the speed data, we can also acquire the corresponding matching road weather, POI, and other external heterogeneous information data. The first section, Sz_speed, records the data from 1 January to 31 January 2015, and has 2976 data points overall. Sz_adj specifies the connectedness of the two roads, with 1 indicating connectivity and 0 indicating non-connectivity, generating a 156*156 adjacency matrix. The second section consists of the diverse data that pertains to the road, such as Sz_Weather and Sz_POI. Five categories of weather conditions are recorded by Sz_Weather: sunny, light rain, heavy rain, cloudy, and foggy. The weather data is likewise gathered at 15-min intervals and, when combined, makes up a 156*2976 matrix. Sz_POI, on the other hand, is the POI information surrounding the road and includes a variety of highly heterogeneous information dimensions. The POI in this paper includes data on restaurants, transportation hubs, educational institutions, healthcare providers, and other services. Each of these services has a corresponding value that indicates the number of POI points that are currently close to the relevant road section, for example, School = 4 indicates that there are 4 schools nearby. This article uses TransE to build the final KG embedding through multi-dimensional external heterogeneous information in the chapter on heterogeneous information embedding. The data set utilized as an input to our model includes the feature matrix that unifies heterogeneous information such as weather and POI.

#### 4.1.2. Baselines

We compare the proposed HIT-GCN with a few earlier models: (1) the Autoregressive Integrated Moving Average model (ARIMA), which makes predictions by fitting a parameter model to the observed time series; (2) the support vector regression model (SVR), a model of a linear kernel function, during which the input and The relationship between outputs; and (3) the T-GCN temporal graph convolutional model. (4) diffusion convolutional



recurrent neural network (DCRNN), (5) the corresponding variant KFDCRNN of DCRNN, (6) Knowledge-Driven Spatial-Temporal graph convolutional network(KF-T-GCN). The hyperparameters of these baseline models are set to be consistent with the hyperparameters of the original papers or released code.

### 4.1.3. Parameter Setup

As a result, we directly specified several parameter values manually for the HIT-GCN model. For instance, we set the learning rate to 0.0005, the batch to 64, the ratio of the training set to the total number of experiments to be 0.8, and the epoch to 5000. The hidden unit size was finally set to 64. The weighting parameters between the similarity of nodes with heterogeneous information and the similarity of the network architecture between nodes, which is also the pair of parameters we focus on, are among the modified parameters $\alpha$ and $\beta$. The weight relationship between the self-representation and the neighborhood-representation is represented by the pair of parameters $\mu$ and $\xi$. We empirically demonstrate that the optimum performance is attained when $\mu = 0.6$ and $\xi = 0.4$. As a result, we explicitly set $\mu = 0.6$ and $\xi = 0.4$ to 0.6 and 0.4 here to examine the effects of varied $\alpha$ and $\beta$ on performance effectively. For all other parameters, we choose the best answer by comparing the model's performance.

### 4.2. Prediction Accuracy Analysis

We compared our approach to the benchmark method with the epoch set to 5000, the number of hidden layer cells set to 64, and all parameters held constant. We depict the velocity measurements from 26 January to 31 January as indicated in Figure 3. To make it simpler to demonstrate the impact of our work, we have provided a comprehensive line graph for 27 January that contrasts our method with the T-GCN and highlights our realistic values in orange. It is clear from the line graph that our HIT-GCN fits the original true velocity values more closely than the original technique, and our prediction model also more closely matches the true values at the line graph's peak regions when the velocities are larger or smaller. This shows that our model performs better than the original T-GCN model and can collect node information more effectively.

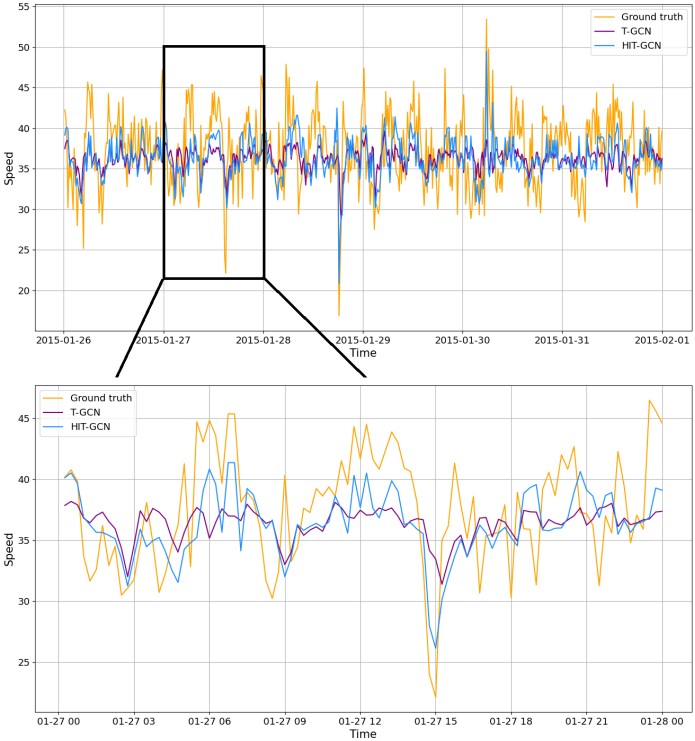

**Figure 3.** Comparison of prediction results of different methods.

### 4.3. Forecast Accuracy, Long-Term Forecast Capability Analysis

We compared our results to a variety of earlier methods that have explored temporal graph convolution models. As shown in Table 1, our HIT-GCN outperformed certain prior approaches in the three elements of Accuracy as well as r2 and var. These three metrics also show the viability and superior performance of our prediction model, because our method concentrates on the heterogeneous information of the nodes and is capable of accurately representing this heterogeneous information. In the feature propagation process, it is possible to take more care of node sections with similar heterogeneous information. And in the model learning process, it also learns the features of nodes with similar heterogeneous information, allowing the model to better predict the speed values of road nodes, i.e., to demonstrate the feasibility of our work.

**Table 1.** Accuracy error comparison chart.

| Evaluation Matrix | SVR | ARIMA | GCN | GRU | DCRNN | KF-DCRNN | T-GCN | KF-T-GCN | HIT-GCN |
|---|---|---|---|---|---|---|---|---|---|
| RMSE | 7.2203 | 6.7708 | 5.6419 | 5.0649 | 4.1243 | 4.0635 | 4.0696 | 4.0443 | **4.0425** |
| MAE | 4.7762 | 4.6656 | 4.2265 | 2.5988 | 2.7514 | 2.7206 | 2.746 | **2.709** | 2.7184 |
| Accuracy | 0.706 | 0.3852 | 0.6119 | 0.7243 | 0.7127 | 0.7169 | 0.7165 | 0.7306 | **0.7312** |
| r2 | 0.8367 | | 0.6678 | 0.8322 | 0.8441 | 0.8487 | 0.8388 | 0.84 | **0.8478** |
| var | 0.8375 | 0.0111 | 0.6679 | 0.8322 | 0.8441 | 0.8491 | 0.8388 | 0.84 | **0.8419** |

Through Table 2, it can be seen that the short- and long-term predictions we made in the long-term speed forecasting were accurate. We did this by conducting four dimensions of prediction comparison trials of 15 min, 30 min, 45 min, and 60 min, respectively. Accuracy and other performance metrics have been significantly improved compared with other prediction models. Our RMSE and MAE results in 45 min and 60 min are nearly identical to those of KF-T-GCN, indicating that the heterogeneous information of the road segment nodes themselves has less influence on the model as time increases in long-term prediction. However, the Accuracy and R2 components are still better than prior work, proving that our model is more effective in short-term or immediate predictions. This implies that both short-term and long-term predicting accuracy is significantly impacted by our model. This illustrates how the model's performance advantages.

**Table 2.** 15 min, 30 min, 45 min, 60 min contrast chart.

| Time | Metric | TGCN | DCRNN | Graph-WaveNet | AST-GCN | KF-T-GCN | HIT-GCN |
|---|---|---|---|---|---|---|---|
| 15 min | RMSE | 4.0696 | 4.5033 | 4.5739 | 4.0294 | 4.0443 | 4.0482 |
| | MAE | 2.746 | 3.17 | 3.218 | 2.7035 | 2.709 | 2.7104 |
| | Accuracy | 0.7165 | 0.7087 | 0.7111 | 0.7193 | 0.7206 | **0.7265** |
| | r2 | 0.8388 | 0.8391 | 0.8422 | 0.8512 | 0.84 | **0.8562** |
| | var | 0.8388 | 0.8391 | 0.8423 | 0.8512 | 0.84 | 0.8418 |
| 30 min | RMSE | 4.077 | 4.5623 | 4.6525 | 4.0529 | 4.0687 | 4.0632 |
| | MAE | 2.747 | 3.23 | 3.2829 | 2.7265 | 2.7228 | 2.7211 |
| | Accuracy | 0.7159 | 0.703 | 0.7047 | 0.7176 | 0.7201 | **0.7221** |
| | r2 | 0.8377 | 0.8332 | 0.8351 | 0.8494 | 0.8372 | **0.8375** |
| | var | 0.8377 | 0.836 | 0.8357 | 0.8495 | 0.8374 | 0.8379 |
| 45 min | RMSE | 4.1035 | 4.6006 | 4.6923 | 4.0822 | 4.0775 | **4.0759** |
| | MAE | 2.7788 | 3.27 | 3.3305 | 2.7611 | 2.7698 | 2.7687 |
| | Accuracy | 0.7141 | 0.6979 | 0.7003 | 0.7156 | 0.7195 | **0.7212** |
| | r2 | 0.8357 | 0.8275 | 0.8301 | 0.8473 | 0.8365 | 0.8448 |
| | var | 0.8357 | 0.8314 | 0.8313 | 0.8474 | 0.8365 | 0.8407 |

**Table 2.** *Cont.*

| Time | Metric | TGCN | DCRNN | Graph-WaveNet | AST-GCN | KF-T-GCN | HIT-GCN |
|------|--------|------|-------|---------------|---------|----------|---------|
| | RMSE | 4.1266 | 4.6412 | 4.7581 | 4.1001 | 4.0798 | 4.0816 |
| | MAE | 2.7911 | 3.31 | 3.3773 | 2.7744 | 2.7768 | 2.7211 |
| 60 min | Accuracy | 0.7125 | 0.6931 | 0.6967 | 0.7143 | 0.7194 | **0.7198** |
| | r2 | 0.8339 | 0.8219 | 0.826 | 0.8459 | 0.8363 | **0.8349** |
| | var | 0.834 | 0.8267 | 0.827 | 0.846 | 0.8364 | 0.8418 |

*4.4. Homophily Adjacency Matrix Analysis*

The main focus of our work is the creation of heterogeneous neighbourhoods matrices through the introduction of external heterogeneous data. Thus, in this section of our experiment, we carry out our experimental work in three different ways to assess the effectiveness of the introduced heterogeneous neighbourhoods matrices on the entire model: (1) analysis of the effect of homogeneous neighbourhoods matrices. We visually analyze the evaluation indicators in the case of adding the homogeneity adjacency matrix and not adding it. (2) Examining the impact of modifying the homogeneous adjacency matrix's parameters on the homogeneous adjacency matrix's parameters, we examine the impact of various parameters on the experiment's results. In this case, our attention is on the impact of the parameters $\alpha$ and $\beta$. (3) Analysis of the association between homogeneous neighbourhood matrices and heterogeneous information in different time dimensions. The effect of our different heterogeneous information-constructed heterogeneous neighbourhood matrices on the experimental effects is analyzed in detail by combining the permutations of heterogeneous information with heterogeneous neighbourhood matrices in different time dimensions. The analysis is also carried out in the time dimension to compare the performance of short-term prediction with that of long-term prediction. In the following, we carry out a detailed description of the three experiments.

4.4.1. Homogeneous Adjacency Matrix Effect Analysis

We measured the effect of introducing the homogeneous adjacency matrix with and without the homogeneous adjacency matrix in each of the two dimensions of RMSE and Accuracy, and the final results are shown in the figure below. Figure 4a shows the rising trend of Accuracy for the first 1000 epochs and convergence at 4000–5000 epochs in Figure 4b. As the number of training rounds increases, the performance of the model is gradually shown, especially in the evaluation metric of Accuracy, the adjacency matrix of heterogeneous information embedding is significant for the prediction performance of the model. Figure 4c contrasts the trend of lowering RMSE for the first 1000 epochs with that of convergence at 4000–5000 epochs in Figure 4d. It is clear that the introduction of the homogeneous adjacency matrix causes the RMSE of the model to decline more quickly than the original adjacency matrix. This is partly because our model pays closer attention to similar neighbouring nodes, allowing the model to learn features more quickly during convergence, potentially improving the accuracy of the model's convergence process. In conclusion, the addition of our homogeneous adjacency matrix has improved the model's convergence and had a positive impact on the accuracy of the final forecast.

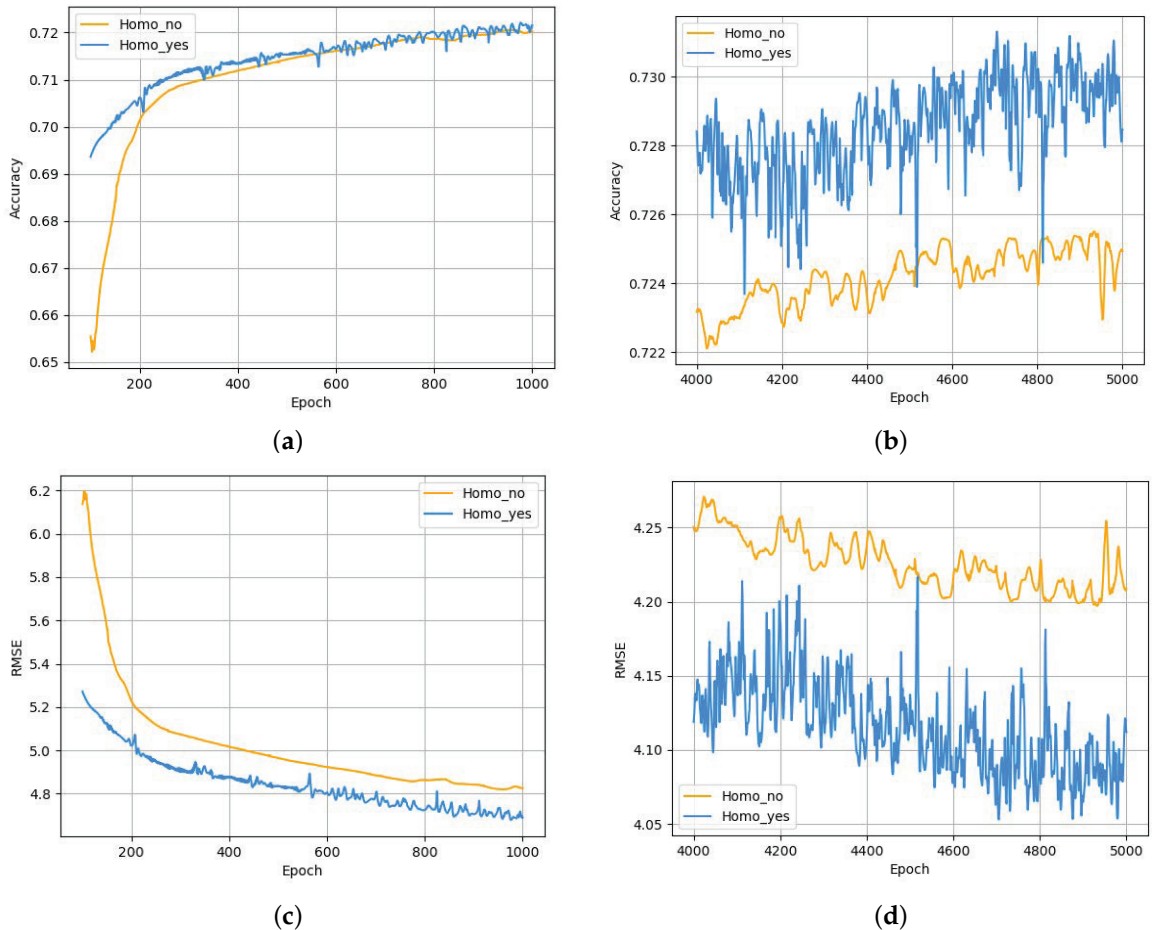

**Figure 4.** Adding homogeneous adjacency matrices. (**a**) Trend chart of Accuracy for the first 1000 epochs. (**b**) Trend chart of Accuracy for the last 1000 epochs. (**c**) Trend chart of RMSE for the first 1000 epochs. (**d**) Trend chart of RMSE for the last 1000 epochs.

4.4.2. Analysis of the Effects of Homogeneous Adjacency Matrix Parameter Adjustment

Figure 5 depicts the final heat map after changes were made for linear combinations of the parameters $\alpha$ and $\beta$. Obviously, different weights have a significant influence on the experiment. The experiment is always poorer in the dimension $\alpha = 0$, regardless of the variance in $\beta$. It is because the situation only analyzes the network's architecture and leaves out the heterogeneous information that the nodes are capable of holding. The combination that most closely matches our mining of heterogeneous information is obtained at $\alpha = 0.4$ and $\beta = 0.8$, where $\alpha$ and $\beta$ are again expressed as the weight parameter between the similarity of nodes with heterogeneous information and the similarity of the network structure between nodes. Additionally, it demonstrates how well our model uses diverse data.

4.4.3. Homogeneous Adjacency Matrix and Heterogeneous Information Correlation Analysis in Long- and Short-Term Forecasting

The four cases of long- and short-term predictive power dynamically combine the embedding dimension of heterogeneous information and the embedding performance of the homogeneous adjacency matrix into a single table, as shown in Figure 6. The case of 15 min, which has heterogeneous information embedded with multiple attributes, achieves the best performance, next-best by the case without the homogeneous adjacency matrix, and the following performance is the case of the longer periods. We have examined the causes of this. Our initial goal in developing the homogeneous adjacency matrix was to increase the performance of the model by adding the adjacency matrix to the heterogeneity of the

external information and to effectively use the information's heterogeneity to improve the model's performance. But in one case alone, the information is not heterogeneous, resulting in the current heterogeneity matrix. Therefore, our model is more suitable for handling multidimensional external information, and the more heterogeneous the external information is, the more significant the performance improvement of the model is, which is in line with our original intention of handling external heterogeneous information and verifies our conjecture. After embedding heterogeneous information, we conducted a comparative examination of the experimental analysis' capacity for long-term prediction. The model's performance in the three dimensions of 30 min, 45 min, and 60 min is consistent with its performance in 15 min, with multidimensional heterogeneous information embedding performing best, followed by the case where such information is not embedded, and finally, the case where single-layer information is still embedded. This shows that the model is stable for long-term prediction while remaining unaffected by the experimental findings. Additionally, in the same dimension, for example, in the case of 45 min, our embedded multiple heterogeneous information performs better than the original adjacency matrix and performs equally well in the case of 60 min, etc. This demonstrates that our model has a similar performance advantage in the ability to make long-term predictions, which also enhances the accuracy of our predictions.

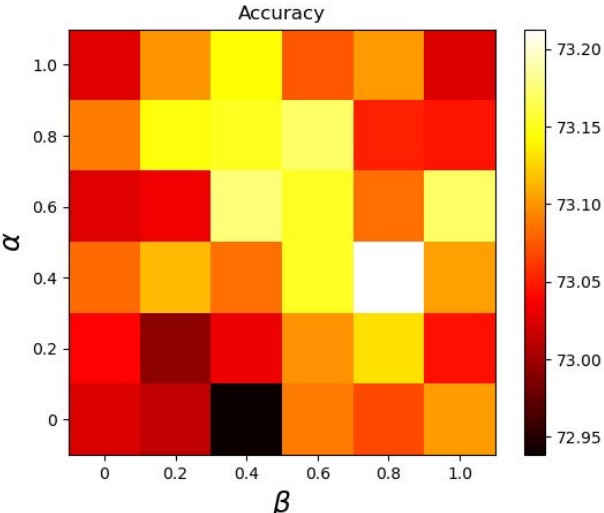

**Figure 5.** homogeneous adjacency matrix parameter adjustment.

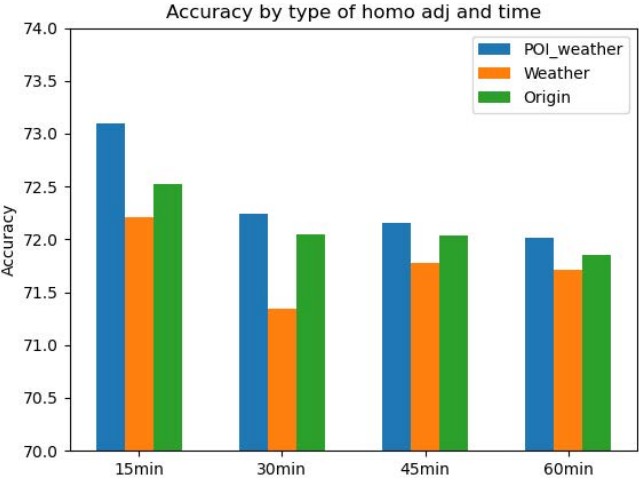

**Figure 6.** heterogeneous information embedding dimension and long-term predictive ability.

### 4.5. Visual Analysis of Prediction Results

Figures 7–10 illustrates our final predictions for the experiment's performance in multiple time dimensions, with our projected values given in red and the actual values in blue over the period of 26 January to 31 January. We also provide a detailed breakdown of the data for the 27th day, as seen in Figure 8. The 15-min prediction results are more closely aligned with the actual outcomes and the two folds fit better, which suggests that our prediction values are more precise, according to our analysis of the visualization findings for the four-time dimensions. With the increase of the time dimension, the fitting degree is not as good as that of 15 min in the 60 min dimension. That is because with the increase of the time dimension, the impact of the time attribute on the prediction performance gradually increases, and the advantage of our model has a downward trend. The phenomenon is also reasonable because the capture performance of the time feature gradually decreases with the increase of the prediction time. It is Given that the impact of temporal attributes on prediction performance gradually increases as the time dimension increases and that our model's advantage tends to decline as the performance in capturing temporal features gradually declines as the prediction time increases, and the fitting becomes worse with the increase of time dimension, up to a dimension of 60 min. The folding line's trend does not gradually reflect the genuine value over time in the detailed graph on the 27th day either. In conclusion, our model's forecasting performance deteriorates over time, but it performs better for short-term forecasts that closely approximate the true value curve.

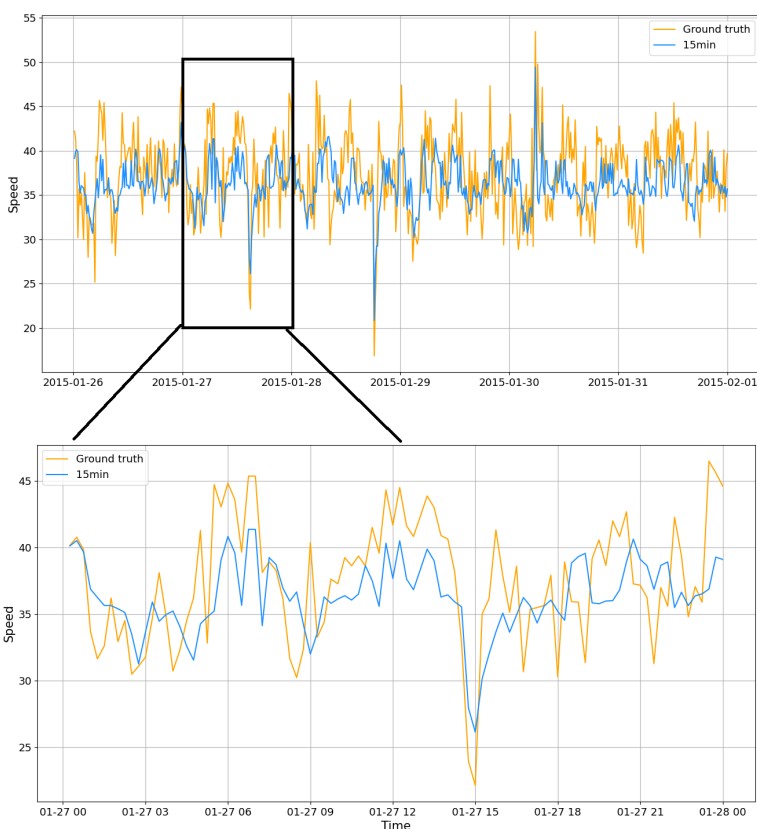

**Figure 7.** Result for prediction horizon of 15 min.

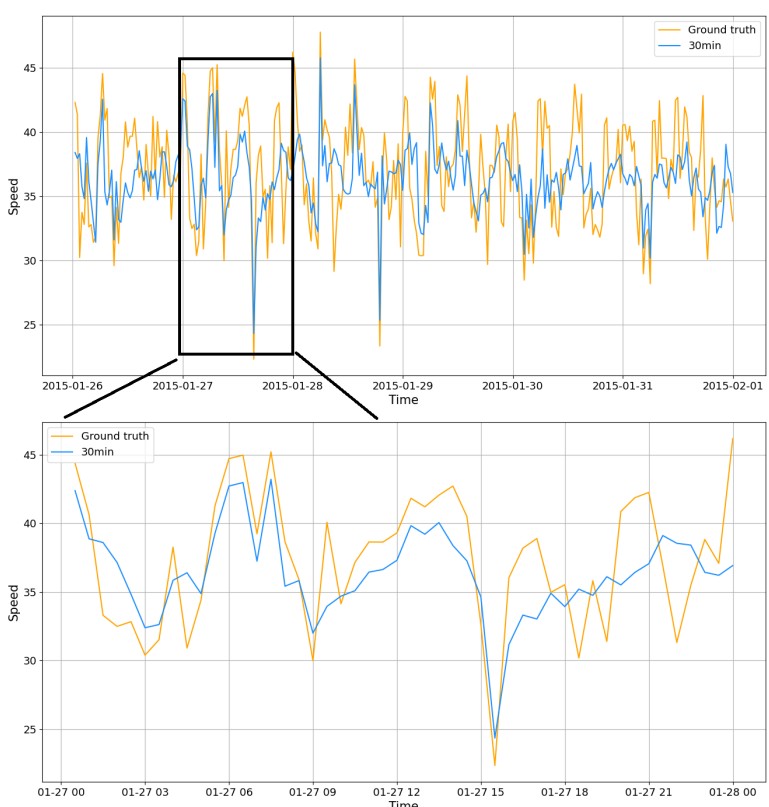

**Figure 8.** Result for prediction horizon of 30 min.

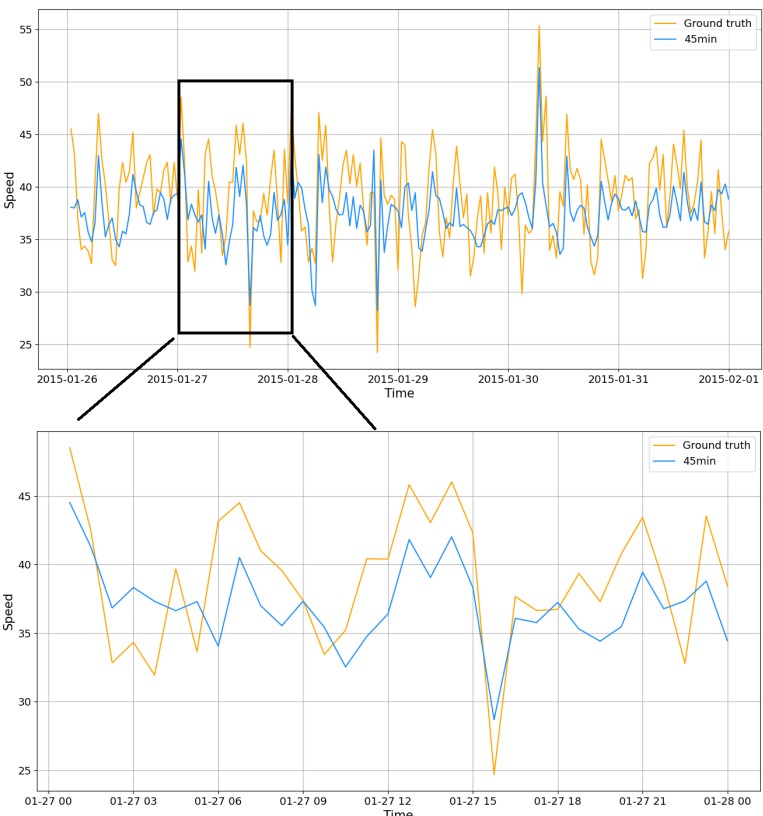

**Figure 9.** Result for prediction horizon of 45 min.

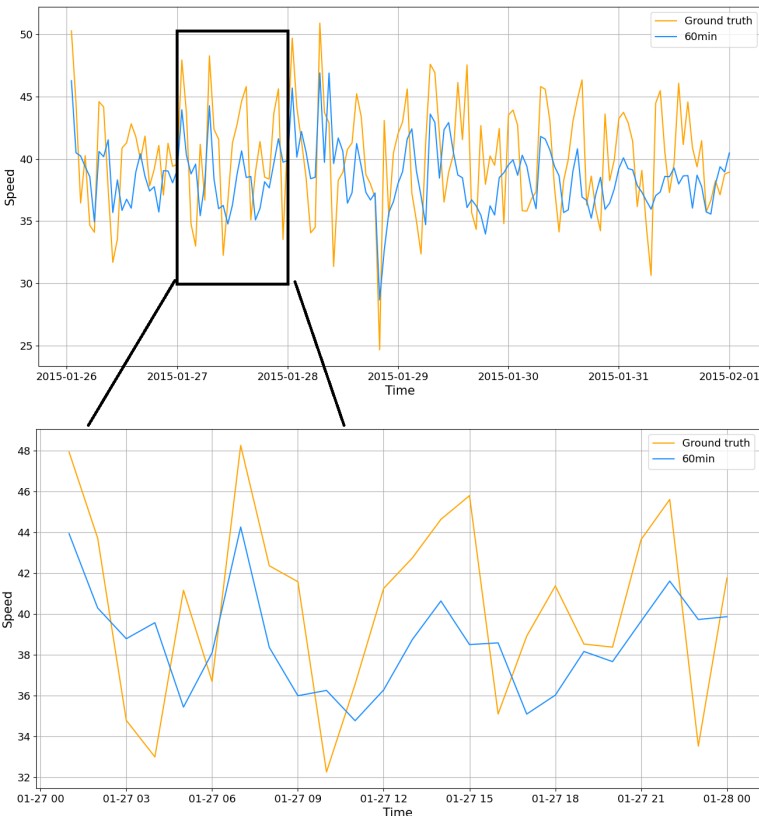

**Figure 10.** Result for prediction horizon of 60 min.

## 5. Conclusions

In our work, we analyzed the heterogeneity among external traffic information, including the existence of one-to-many and many-to-one relationships. To address this issue, we proposed a spatial-temporal graph convolutional network embedded with heterogeneous information of road network (HIT-GCN). We utilized the degree of heterogeneity between nodes to modify the network propagation, and investigated the impact of this modification on the performance of the prediction model. Specifically, we first generated knowledge vectors to quantify the heterogeneity among node attribute information by embedding external heterogeneous information at the individual node level. Then, at the road network level, we calculated a homogeneous adjacency matrix based on both the topological structure of the road network and the similarity in heterogeneity between nodes. We used this matrix to determine whether two road segment nodes possess similar spatial features based on their topological structure and similarity in heterogeneous information, and assigned different weights to neighboring nodes to guide the propagation of the spatiotemporal graph convolutional network and improve the accuracy and reliability of the prediction. Experimental results on a real-world dataset showed that our model significantly outperformed previous models that only dealt with homogeneous or single-dimensional information in both short-term and long-term predictions when the homogeneous adjacency matrix was introduced into our model. Additionally, sensitivity analyses demonstrated the significant impact of different embedding dimensions and parameter settings on experimental results. Despite this, our work still demonstrated superior performance compared to baseline models.In our work, the homogeneous adjacency matrix is calculated by combining the heterogeneous information of the nodes with the node features of the road network through knowledge graph embedding. Additionally, the topology of the road network is taken into account, as it directs the propagation of the feature extraction in GCN, allowing for the treatment of nodes with similar heterogeneous information or network topology. Our results demonstrate that the homogeneous adjacency matrix is effective in handling hetero-

geneous information, and the ablation experiments also demonstrate that the dimension of the embedded heterogeneous information significantly affects the experimental results. The homogeneous adjacency matrix also improves the prediction accuracy of speed and long-term prediction ability.

The effective utilization of heterogeneous attribute information has great application value in the field of traffic prediction. In practical intelligent transportation systems, road segments often carry multidimensional information, including numerous instances of heterogeneous or homogeneous information. It remains a challenge to efficiently integrate this information. In our future work, we will explore two aspects of utilizing heterogeneous information: the dimensions of such information and downstream tasks, such as travel demand estimation, in speed prediction. Only by doing so can we better meet the needs of transportation systems in real-world settings.

**Author Contributions:** Conceptualization, H.X., G.S. and H.Y.; methodology, H.X., X.L. and X.K.; software, H.X.; formal analysis, H.X. and X.L.; investigation, H.X.; data curation, H.X.; writing—original draft preparation, H.X. and H.Y.; writing—review and editing, G.S. and X.K.; funding acquisition, G.S. and X.K. All authors have read and agreed to the published version of the manuscript.

**Funding:** This work was supported in part by the "Pioneer" and "Leading Goose" R&D Program of Zhejiang under Grant 2022C01050, in part by the Zhejiang Provincial Natural Science Foundation under Grant LR21F020003, and in part by the National Natural Science Foundation of China under Grant 62072409 and Grant 62073295.

**Data Availability Statement:** The data presented in this study are available on request from the corresponding author. The data are not publicly available due to privacy.

**Conflicts of Interest:** The authors declare no conflict of interest.

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
