# Peer review of "HIT-GCN: Spatial-Temporal Graph Convolutional Network Embedded with Heterogeneous Information of Road Network for Traffic Forecasting"

_electronics, doi:10.3390/electronics12061306_

Round 1

Reviewer 1 Report

The current paper addresses an important issue that occurs almost in every big cities around the world – traffic. It is known that the number of vehicles has increased over the years, and this leads to increased time in traffic and also causes traffic congestion. The authors propose an introduction of a 14-adjacency matrix, that can improve the prediction of the traffic speed, using graph convolutional networks. Moreover, the authors have demonstrated the benefits of heterogeneous features in traffic forecasting. Overall, this paper it is a great contribution for traffic prediction.

Author Response

Dear reviewers, thank you very much for your valuable comments on our manuscript. There is no doubt that these comments are valuable and helpful for revising and improving our manuscript. We thank Reviewer 1 again for the constructive comments and suggestions which have helped us to enhance the quality of this paper.

Reviewer 2 Report

This paper presents a heterogeneous approach in predicting traffic conditions, taking into account external factors (weather conditions, the existence of traffic stations, emergencies, holidays, and the distribution of nearby POIs, etc.). To handle heterogeneous information, the authors propose to use a convolutional network.

This paper is presented in a very interesting and easy-to-read manner. However, there are a few comments which might be confirmed:

1. The main problem with why the authors propose to take advantage of traffic heterogeneity is not clear. This needs to be explained in detail in the abstract and background.

2. Previous work related to traffic prediction based on heterogeneous graphs has been carried out a lot. Even though the author conveys the update, the gap between the proposed system and the existing system is not clear.

3. Can the author confirm that the number of datasets used is sufficient? Dataset January 1 to January 31, 2015, while there are weather features (related to seasons).

4. The author should also convey the parameter settings for the model baselines. does this come from preliminary research or reference?

5. It is important for the reader that the paper presents a few examples of prediction error calculations

Author Response

Dear reviewers, thank you very much for your valuable comments on our manuscript. Below, we carefully revised the manuscript based on the reviewers’ comments and suggestions. Please see the attachment.

Reviewer 3 Report

Review report

The processing performed by the authors properly collapsed the data, resulting in a very high level of predictive accuracy. Comparisons with other methods were also made, and one does not have to be a specialist in the field to understand that the methods devised by the authors are effective. I felt that improving accuracy by multiplying multiple parameters by a single parameter would be a promising basic method for the future in this field. Therefore, we believe that the proposal deserves acceptance if some of the comments are addressed.

Comments

1.     The font size of the drawings is not consistent. Also, the resolution of the images from Fig. 7 to Fig. 10 is low.

2.     The format of some of the references is incorrect. Some years are in bold and some in normal format.

3.     The font is different in Table 2 compare with other part.

4.     In line 435 to 436, what is the meaning of previous work? It is not clear.

5.     In line 436 to 437, authors described KST-GCN. This might be KF-T-GCN.

6.     For Figure 4, four are listed from (a)-(d), but there is no explanation in the graph and it is unclear.

Author Response

Dear reviewers, thank you very much for your valuable comments on our manuscript.Below, we carefully revised the manuscript based on the reviewers’ comments and suggestions.Please see the attachment.

Round 2

Reviewer 2 Report

This paper is presented better than the previous paper. However, there is a minor suggestion that makes this paper easier to read:

It is better if the authors show the accuracy of the proposed method in the abstract and conclusion.
